# Self-organized reactivation maintains and reinforces memories despite synaptic turnover

Michael Jan Fauth[1,2], Mark CW van Rossum[3,4]*

[1]School of Informatics, University of Edinburgh, Edinburgh, United Kingdom; [2]Third Physics Institute, University of Göttingen, Göttingen, Germany; [3]School of Psychology, University of Nottingham, Nottingham, United Kingdom; [4]School of Mathematical Sciences, University of Nottingham, Nottingham, United Kingdom

**Abstract** Long-term memories are believed to be stored in the synapses of cortical neuronal networks. However, recent experiments report continuous creation and removal of cortical synapses, which raises the question how memories can survive on such a variable substrate. Here, we study the formation and retention of associative memory in a computational model based on Hebbian cell assemblies in the presence of both synaptic and structural plasticity. During rest periods, such as may occur during sleep, the assemblies reactivate spontaneously, reinforcing memories against ongoing synapse removal and replacement. Brief daily reactivations during rest-periods suffice to not only maintain the assemblies, but even strengthen them, and improve pattern completion, consistent with offline memory gains observed experimentally. While the connectivity inside memory representations is strengthened during rest phases, connections in the rest of the network decay and vanish thus reconciling apparently conflicting hypotheses of the influence of sleep on cortical connectivity.
DOI: https://doi.org/10.7554/eLife.43717.001

*For correspondence:
mark.vanrossum@nottingham.ac.
uk

## Introduction

Long-term memories are believed to be stored in the connections of cortical neuronal networks (*Martin et al., 2000*; *Mayford et al., 2012*). While it is often assumed that the synaptic connectivity remains stable after memory formation, there is an increasing body of evidence that connectivity changes substantially on a daily basis. Continuous rewiring of the synaptic connections (*Holtmaat et al., 2005*; *Xu et al., 2009*; *Yang et al., 2009*; *Loewenstein et al., 2015*) may exchange up to 50% of the synapse population over a time-course of weeks (*Loewenstein et al., 2015*). Hence, only a minuscule fraction of synaptic connections generated upon the original learning experience persists after a few months (*Yang et al., 2009*). Intriguingly, experiments demonstrate that despite this continuous synaptic turnover, memories are not only stable but might even improve without further training (*Walker et al., 2003*; *Cai and Rickard, 2009*; *Honma et al., 2015*), especially during sleep (*Jenkins and Dallenbach, 1924*; *Karni et al., 1994*; *Fischer et al., 2002*; *Walker et al., 2003*; *Dudai, 2004*; *Stickgold, 2005*; *Gais et al., 2006*; *Korman et al., 2007*; *Lahl et al., 2008*; *Diekelmann and Born, 2010*; *Pan and Rickard, 2015*; *Rickard and Pan, 2017*).

It is unclear how cortical networks can retain, let alone improve, memories over time (*Mongillo et al., 2017*; *Ziv and Brenner, 2018*; *Rumpel and Triesch, 2016*). One hypothesis is that there are two pools of synapses: a stable and an unstable one (*Kasai et al., 2003*; *Loewenstein et al., 2015*), and in particular inhibitory stability could play a crucial role (*Mongillo et al., 2018*). Another possibility is that, by hippocampal coordination, memory events are replayed in the cortex. For example, *Acker et al. (2018)* show that periodic, external replay of

learned input patterns strengthens synaptic connections that are consistent with existing connections. However, as the case of patient H.M. shows, hippocampal replay does not appear to be necessary to maintain cortical memories. Finally, in *Kappel et al. (2018)*, synaptic plasticity depending on a global reinforcement signal is used to stabilize new synapses which contribute to a learned behavior. However, such a reinforcement signal seems biologically implausible to maintain long-term memories.

In this study, we will explore another possibility, namely that the Hebbian cell assemblies that store the memories (*Hebb, 1949*; *Palm, 1982*; *Harris, 2012*; *Palm et al., 2014*; *Litwin-Kumar and Doiron, 2014*), spontaneously transiently reactivate. Such spontaneous alternation between high and low population activity has previously been associated with Up-Down-dynamics (*Holcman and Tsodyks, 2006*; *Barak and Tsodyks, 2007*; *Setareh et al., 2017*; *Jercog et al., 2017*), observed in cortical networks during sleep (*Steriade et al., 1993*) and in quiescent awake states (*Poulet and Petersen, 2008*; *Okun et al., 2010*; *Engel et al., 2016*). Along these lines, experiments show that learning- and memory-related activity patterns are reactivated in cortex, predominantly during sleep and rest (*Peyrache et al., 2009*; *Gulati et al., 2014*; *Ramanathan et al., 2015*; *Gulati et al., 2017*; *Jiang et al., 2017*). Recent modeling work has shown that in the absence of synaptic turnover, reactivation can indeed maintain stable memories in recurrent networks (*Tetzlaff et al., 2013*; *Litwin-Kumar and Doiron, 2014*; *Zenke et al., 2015*). However, it is unclear whether memories can be made robust against synaptic turnover which would be necessary to account for long-memory in biological networks.

Synaptic turnover partly depends on neuronal activity or the size of the synapses (see *Fauth and Tetzlaff, 2016* for a review). In particular, larger synapses are less likely to be removed (e.g. *Le Bé and Markram, 2006*) implying that rewiring follows synaptic plasticity and attains Hebbian-like character. Here, we show that the combination of structural plasticity, synaptic plasticity and self-generated reactivation, even for a just short period every day, can not only stabilize assemblies against synaptic turnover but even enhance their connectivity and associative memory.

## Results

Using a computational model, we investigate the storage and long-term stability of memories in the presence of short-term depression and the two major activity-dependent processes that are thought to implement long-lasting cortical connectivity changes (*Figure 1A*, for a review see *Fauth and Tetzlaff, 2016*): (1) Synaptic plasticity, which changes the transmission efficacy - or synaptic weight - between neurons (see *Martin et al., 2000*; *Takeuchi et al., 2014* for a review), and (2) structural plasticity, that is, the creation and removal of synapses. Structural plasticity is strongly correlated to

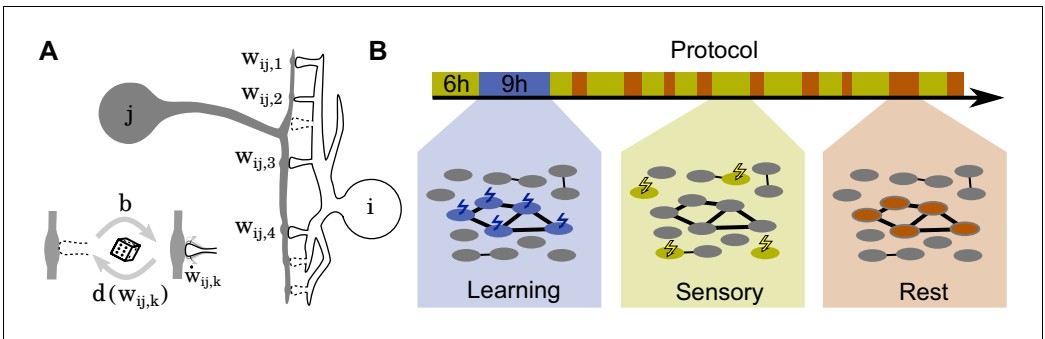

**Figure 1.** Schematics of model and stimulation paradigm. (A) Neuron $j$ is connected to neuron $i$ at $S_{max}$ potential synaptic locations (here $S_{max} = 7$). (Inset) Non-functional potential synapses (dashed) become functional synapses (solid) with a constant rate $b$ and are deleted with a weight-dependent rate $d(w_{ij,k})$. The weights of functional synapses are adapted by a Hebbian plasticity rule $\dot{w}_{ij,k}$. (B) The simulation protocol is structured in three phases: During a learning phase (blue) groups of neurons are strongly stimulated in an alternating fashion. During subsequent sensory phases (yellow), the network is bombarded with quickly changing patterns. Finally, during rest phases (red), the network receives no stimulation but spontaneously reactivates.
DOI: https://doi.org/10.7554/eLife.43717.002

successful learning (*Patel and Stewart, 1988*; *Patel et al., 1988*; *Kleim et al., 2002*; *Xu et al., 2009*; *Yang et al., 2009*; *Lai et al., 2012*; *Moczulska et al., 2013*), but also gives rise to continuous synaptic turnover. We study how these plasticity processes store co-activity patterns in the connectivity of recurrent neuronal networks and how the corresponding connectivity changes are retained and strengthened over time.

The simulation protocol has three phases (*Figure 1B*): During the **learning phase**, alternate groups of neurons receive strong stimulation and the patterns are stored into the network. Next, during the **sensory phase**, the network receives ongoing input from upstream networks mimicking incoming sensory information. Finally, during the **rest phase**, the neurons do not receive any external stimulation but re-activate spontaneously.

## Assembly formation during learning phase

As a first step to investigate memory storage in the presence of synaptic and structural plasticity, we examine how cell assemblies are formed. We simulate a learning phase where multiple groups of neurons receive strong, external stimulation in an alternating fashion. This stimulation leads to high activity in these neurons, and low activity in the rest of the network due to lateral inhibition (*Figure 2A*).

We track the time-course of the average synaptic weight (*Figure 2B*, Top) and the average number of synapses (*Figure 2B*, Bottom) in three different classes of connections: First, connections within the same stimulated group (intra-assembly connections, purple curves) are potentiated as a result of intervals of correlated high pre- and postsynaptic activity. As the decay occurring between these intervals is limited, there is a fast increase of the weights (*Figure 2B*, Top, purple curves). On a slower time-scale, new connections are build up (*Figure 2B*, Bottom). The build up occurs because, while the connection creation rate is constant, the deletion probability decreases for larger weights, in line with experiments (*Le Bé and Markram, 2006*; *Yasumatsu et al., 2008*).

Second, neurons outside the assembly, also referred to as control neurons, are not stimulated and exhibit low activity. Connections between them are not potentiated and remain unstable, leading to a low connectivity between these neurons (*Figure 2B*, grey curves). Third, connections between different stimulated groups as well as connections between control neurons and stimulated

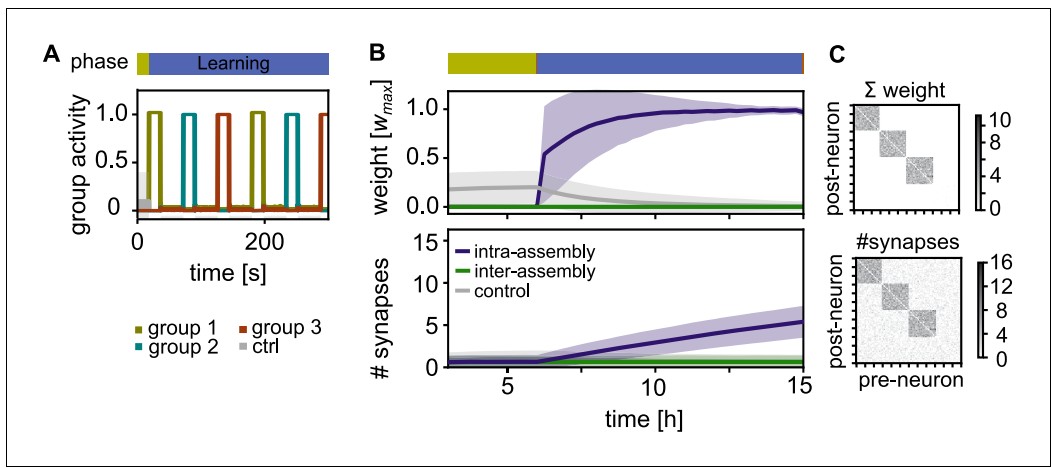

**Figure 2.** Activity and connectivity during the learning phase. (**A**) Activity during the learning phase when assemblies receive strong, alternating stimulation. (**B**) Top: Time-course of the synaptic weights per synapse during learning. Between neurons within an assembly (purple), between control neurons (grey), and between different assemblies or assembly and control neurons (green). Curves depict the mean and shadings the standard deviation. Bottom: The slower time-course of the number of synapses per connection. (**C**) Top: Matrix depicting the sum of weights between 120 exemplary neurons (including all three stimulated groups) after learning at $t = 15\text{h}$. Each point represents the synaptic weight summed over all the synaptic connections between the two neurons (x-axis pre- and y-axis-postsynaptic neuron; sorted). Bottom: Corresponding connectivity matrix of the number of synapses.

DOI: https://doi.org/10.7554/eLife.43717.003

groups are asynchronously active. These connections have small weights, such that synapses are unstable and the number of synapses remains low (*Figure 2B*, green curves).

In summary, during the learning phase, each stimulated group becomes a highly interconnected cell assembly, while all non-intra-assembly synapses remain sparse and have small synaptic weights (*Figure 2B*). The synaptic connectivity follows the synaptic strength on a time-scale of hours.

## Cell assemblies spontaneously reactivate

After the learning phase, the network alternates between two modes. During sensory phases, the network receives stimulation with quickly changing random patterns mimicking incoming information from upstream networks. In this phase, lateral inhibition prevents spontaneous activation of non-stimulated neurons and the assemblies slowly decay.

However, during rest phases, stimulation is absent and the assemblies can reactivate. This happens provided there is sufficient intra-assembly recurrence, so that the positive feedback drives the assemblies towards a high population activity (e.g. *Wilson and Cowan, 1972*; *Brunel, 2000*; *Figure 3A*, black nullcline), which corresponds to a reactivation of the learned pattern. As lateral inhibition implements a winner-take-all structure, only one of the assemblies reactivates at any time.

To confirm this, we tracked the pre- and postsynaptic activities for all connections during the rest phase. For connections within the same assembly, the activity is strongly correlated (*Figure 3Ci*). We find a high probability of experiencing either high activities in both pre- and postsynaptic neuron (self-reactivations) or low activities (activation of other assemblies). In contrast, for connections between pairs of neurons from different assemblies or from assembly and control group there is virtually no correlated activity (*Figure 3Cii*).

The high activity state is sustained by strong positive feedback from inside the assembly. To prevent that the assembly stays highly active, the positive feedback-loop must be shut down. Similar to other models (*Barak and Tsodyks, 2007*; *Holcman and Tsodyks, 2006*), short-term depression weakens the transmission efficacy of the excitatory synapses and thereby the positive feedback. As a consequence, the high population activity becomes unstable (i.e. the high activity fixed-point vanishes, *Figure 3A*, grey nullcline) such that the activity drops back and the synapses can recover.

## Self-reactivation strengthens cell assemblies

Next, we investigate how reactivation is crucial for maintaining the assemblies. If there are no rest phases in which the assemblies can reactivate, the synaptic weights inside the assemblies decay

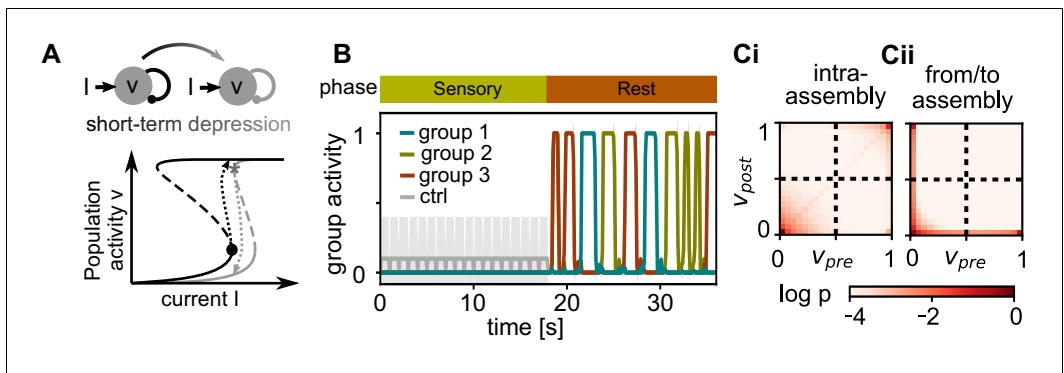

**Figure 3.** Activity after learning. (**A**) Sketch of the dynamics during the rest phase. Solid lines mark the fixed points of population activity in a strongly connected assembly for relaxed weights (black) and after short-term depression (grey). When the assembly is taken beyond a upper bifurcation point (black dot), it becomes highly active (black dashed arrow). Subsequently, short-term depression decreases the recurrent connectivity and the population activity falls back to the low activity (grey dashed arrow). (**B**) Mean activity in stimulated groups and control neurons during sensory and rest phase. Shadings depict standard deviations. (**C**) Correlation of pre- and postsynaptic activity during the rest phase for intra-assembly connections. (**Cii**) Same for connections between two stimulated groups and between stimulated groups and control neurons.

DOI: https://doi.org/10.7554/eLife.43717.004

(*Figure 4A*, Top). Because the synaptic removal rate is faster for small synapses, the synaptic weight decay is followed by a decay in the number of synapses on a timescale of days (*Figure 4A*, Bottom).

In contrast, when the sensory phases are interleaved with rest phases, we observe that the strong connectivity inside the assembly is not only preserved by the self-reactivations, but even gradually increases (*Figure 4B*, Bottom, purple curve). Moreover, the connectivity from and to other assemblies in the network remains weak (*Figure 4C*). Close inspection of the connectivity changes reveals that also spurious connections that were built-up during the sensory phases are removed during the rest phase (*Figure 4—figure supplement 1*).

In the absence of structural plasticity, maintaining the memories is more challenging. In *Figure 4D*, we blocked structural plasticity during the retention phase. The assemblies initially reactivate and are stable for multiple days, but a long sensory phase after 100 hours drives them below the reactivation threshold and all assemblies are lost.

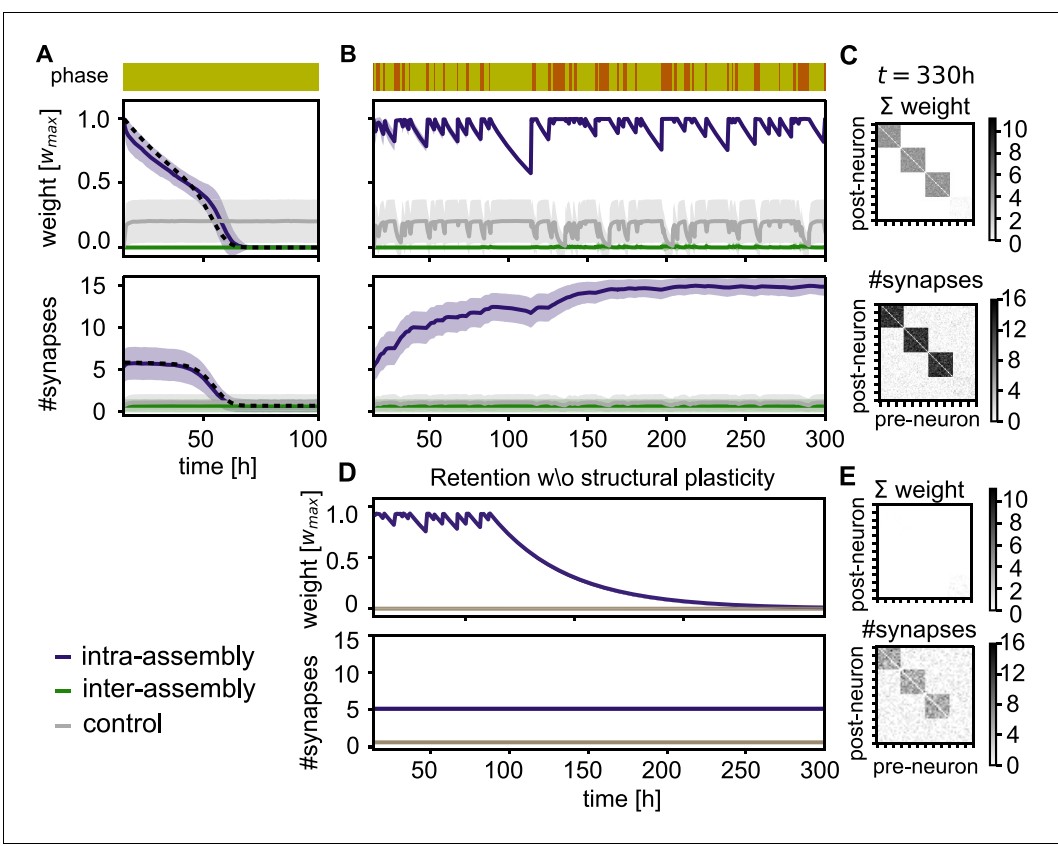

**Figure 4.** Stabilization of connectivity after learning. (**A**) Without resting intervals mean intra-group weights decay (purple), followed by a decay in connectivity (bottom panel). Weights between control neurons (grey) and between different assemblies or assembly and control neurons (green) remain low. Dashed line depicts theoretical prediction; shading represents standard deviation. (**B**) Reactivation during rest phases rescues the assemblies. (**C**) Connectivity matrix depicting the sum of weights (top) and number of synapses (bottom) between 120 exemplary neurons at the end of the simulation in panel B. (**D**) Simulation with the same sequence of sensory and rest phases, but all structural plasticity is blocked after learning ($t = 15h$). The memory is retained for a while, but after about 100 hr, when there is a longer period without reactivation, the assemblies decay. (**E**) Connectivity matrices as in panel C for the simulation in panel D. Although the number of synapses is similar, the synaptic weights have decayed.

DOI: https://doi.org/10.7554/eLife.43717.005

The following figure supplements are available for figure 4:

**Figure supplement 1.** Connectivity changes in individual phases.
DOI: https://doi.org/10.7554/eLife.43717.006
**Figure supplement 2.** Net weight change after 24 hr of repeated sensory and rest phases.
DOI: https://doi.org/10.7554/eLife.43717.007

## Synapses in assemblies are continuously replaced

The assembly strengthening during the rest phase could be due to addition of new synapses, or the net result of concurrent creation and removal. Similar to experiments (*Holtmaat et al., 2005*; *Yang et al., 2009*; *Loewenstein et al., 2015*), we measured the persistence of synapses at a specific potential synaptic locations from day to day and find ongoing synapse creation and removal within the assemblies, exchanging around 10% of the synapse population on a daily basis (*Figure 5A*). Moreover, synapse creation transiently increases shortly after learning while spine removal remains low, similar as in experiments (*Yang et al., 2014*). The fraction of synapses that were present after learning and persist until a given time-point continuously decreases (*Figure 5B*) indicating that also strong synapses inside the assemblies are continuously removed. Hence, the strengthening of the assemblies is not emerging from a simple addition of new synapses, but because the reactivations stabilize more synapses than are being removed and thus the synaptic substrate of the assemblies is continuously exchanged (*Figure 5C*).

## Self-reinforcements improves robustness of pattern completion

Knowing that cell assemblies can strengthen their connectivity by self-organized reactivation, we examine whether this improves the associative properties of the encoded memories. Associative memory requires the re-activation of an assembly, even when presented with a cue that only partially overlaps with the memory.

Quick and correct recall of a cued pattern requires that those neurons which should be active in this pattern receive a strong input and those neurons which should stay silent receive a weak input. The further the input distributions are separated from each other and from the offset of the neuronal gain function, the better and faster the recall quality. We investigate the retrieval robustness of the memory against corruption in the cue, by switching a fraction of the active neurons of the pattern off and the same number of inactive neurons on. Immediately after learning ($t = 15\text{h}$, left most time-point in 4B), the input distributions partly overlap (*Figure 6A*). However, after reactivations during rest ($t = 300\text{h}$), the distributions are separated further and the currents of the active neurons are above the offset of the sigmoidal gain function (*Figure 6B*), such that the pattern will be correctly completed.

To quantify this further, we evaluated the quality of pattern completion for varying levels of pattern corruption. Directly after learning, errors increase when the corruption exceeds 15% (*Figure 6C*), mostly due to an increase in false negatives (*Figure 6D*). After the resting phases, the

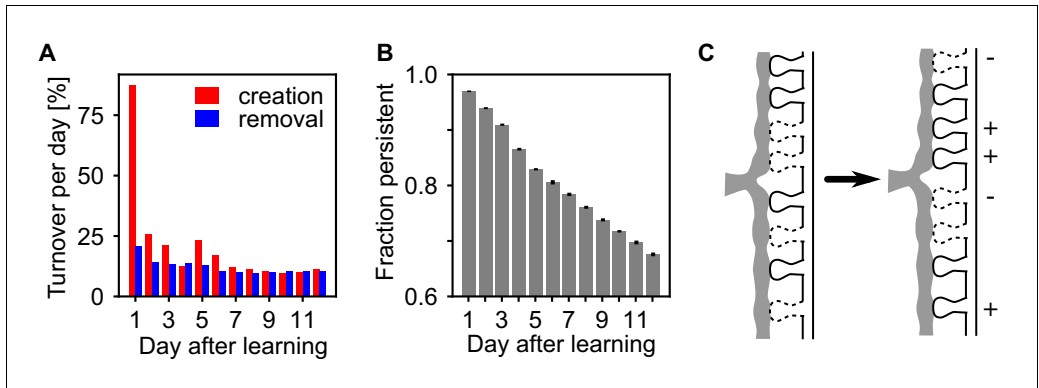

**Figure 5.** Assemblies undergo strong synapse turnover. (**A**) Synapse creation (red) and removal (blue) relative to the synapses existing at the previous day. Synapse creation is strongly elevated the first day after learning. (**B**) The decay of persistent synapses created during learning indicate ongoing removal and replacement of synapses that originally formed the memory. (**C**) Sketch of the structural changes occurring in the presence of self-reactivation. Although the synapses that have been created during learning are continuously removed (-), the collective dynamics of the assembly stabilizes new synapses (+) between its neurons which counteracts synapse loss and can lead to a net strengthening of the assembly.

DOI: https://doi.org/10.7554/eLife.43717.008

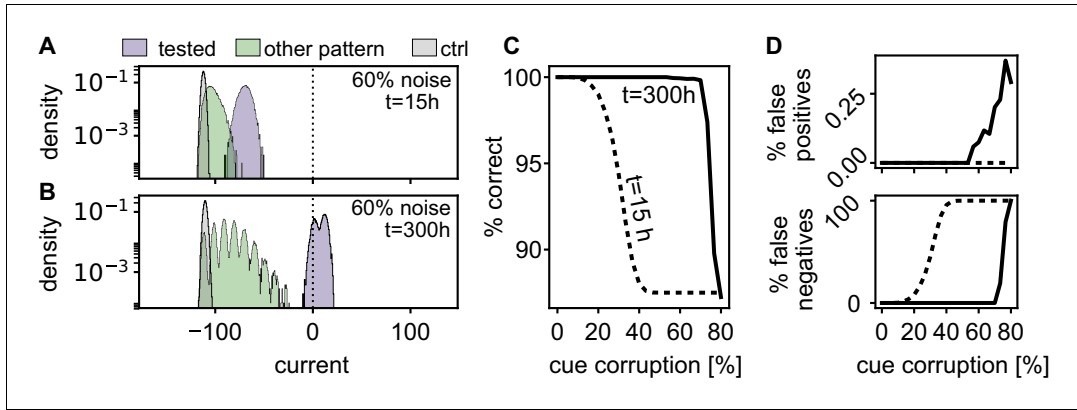

**Figure 6.** Spontaneous reactivation improves robustness of pattern completion. (**A**) Histograms of the incoming currents to different classes of neurons right after learning ($t = 15\mathrm{h}$), when presented with corrupted memory patterns (histograms from 100 realizations). For successful pattern completion, the incoming currents of neurons in the tested stimulation pattern (blue) should be above the threshold at zero while the currents from neurons that are part of another pattern (green) and neurons that have not been stimulated during learning (grey) should be below. (**B**) After the reactivations during rest phases, the distributions are more separated ($t = 300\mathrm{h}$). (**C**) Fraction of correctly classified neurons (active and inactive) for different levels of cue corruption. (**D**) Fraction of wrongly inactive (bottom) and wrongly active neurons (top).
DOI: https://doi.org/10.7554/eLife.43717.009

strengthened network ($t = 300\mathrm{h}$) recalls correctly up till much higher noise levels. In summary, the self-induced strengthening improves recall quality.

## Timescale requirements for assembly strengthening

Given that spontaneous reactivation during the rest phase is thus crucial for memory maintenance (*Figure 4B*), we analyze how frequent and how long the rest phases need to be. Starting from a given number of synapses between the neurons in the assembly, we find that the number of connections increases more the longer the rest phases and the shorter sensory phases (*Figure 7A*). When the sensory phases are too long ($\tau_{rnd} > 20$ hr), the assemblies loose synapses (blue region).

This behaviour can be described theoretically (Materials and methods): We assume that all synapses in an assembly are maximally potentiated by reactivation during the rest phase and decay exponentially during sensory phases and then calculate the expected synaptic creation and removal. This simple theory explains the simulation results over a broad regime of timescales (*Figure 7B*) indicating that the dynamics of the intra-assembly synapses is governed by the collective reactivation of the assemblies.

It is also possible, however, that assemblies fail to reactivate and decay instead. To quantify when such reactivation failures occur, we evaluated the maximal duration of the sensory phase after which at least 90% of assemblies still reactivate for different initial connectivity levels. Unsurprisingly, the larger the number of initial connections, the longer the assemblies survive (*Figure 7C*). This, in turn, implies that by increasing their connectivity by self-reactivation, the assemblies also become more and more stable against prolonged absence of reactivation.

## Strengthening emerges from convergence to an attractive state

Connections in an anssembly can undergo two fates: either the connections decay to control levels or they converge to a fixed number of synapses (*Figure 7D*). Self-organized strengthening, as observed above, occurs when the initial number of synapses is below the fixed point, yet the assembly is strong enough to survive and self-reactivate (e.g. traces starting at $S = 8$ or $S = 12$ in *Figure 7D*).

To study the emergence of this fixed point in the number of synapses, we simulated the change in the number of synapses after one full cycle of sensory and rest phase (*Figure 7E*). The number of synapses either decreases for large initial numbers of synapses and increases for small initial

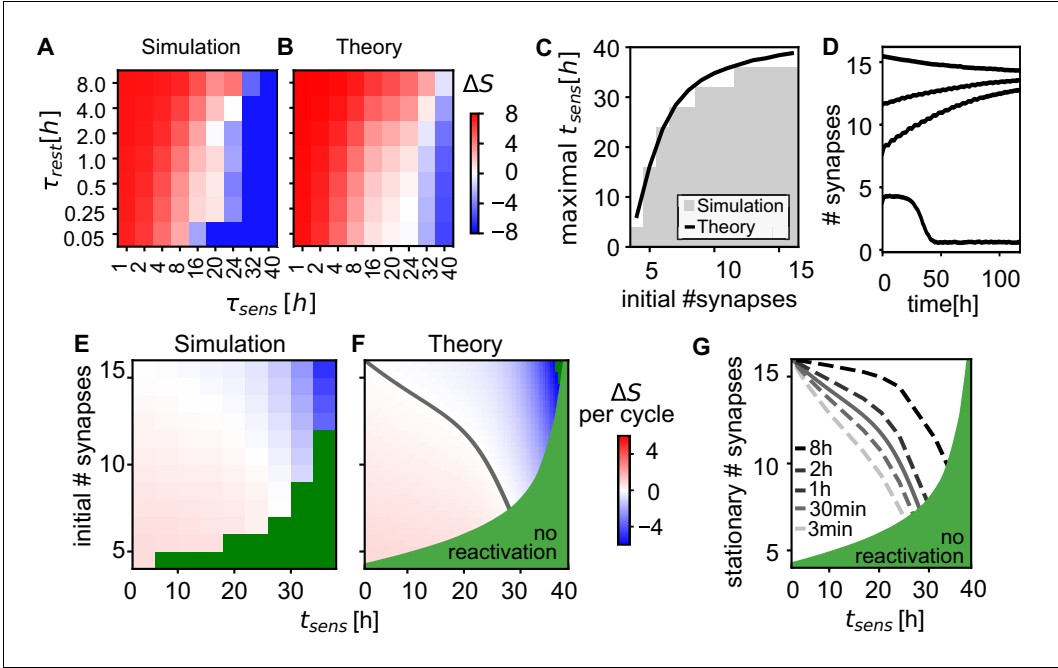

**Figure 7.** Analysis of connectivity decay and stationary weights. (**A**) Change of the number of synapses between intra-assembly neurons after 5 days. The intra-assembly connectivity grows stronger for longer rest phases and shorter sensory phases. For long sensory phases connectivity decreases. (Averaged over fifteen 30-neuron assemblies initialized with eight synapses per connection.) (**B**) The theoretical prediction matches the simulation. (**C**) Maximal duration of sensory phase after which assemblies still self-reactivate increases when starting of with a larger initial numbers of synapses. The curve shows the theoretical predictions of the latest possible reactivation. (**D**) Time-course of the average number of synapses (network alternating between $t_{sens} = 4.5h$ sensory phases and $t_{rest} = 0.5h$ rest phases). Above a minimal initial number of synapses, a convergence to a stable state can be observed. (**E**) Change in the number of synapses after one cycle of sensory and rest phase for different initial numbers of synapses and durations of the sensory phase ($t_{rest} = 1\text{h}$). In the green region, reactivations will not occur (see panel C) and assemblies will decay (see panel D). (**F**) Theoretical prediction of the change per cycle (color code) matches experiment and exhibits a stable stationary value (solid grey curve). (**G**) Theoretically predicted stationary number of synapses for different rest phase durations.
DOI: https://doi.org/10.7554/eLife.43717.010

numbers of synapses, matching theory (*Figure 7F*). Hence, after many cycles with the same phase durations, the number of synapses will converge and fluctuate around a stationary final value (gray line). The theory provides the dependence of this final value on the durations of sensory and rest phases: The longer the rest phase and the shorter the sensory the higher the value (*Figure 7G*).

For long sensory phases (x-axis in *Figure 7E,F and G*), the synaptic survival probability diminishes. The stationary number qualitatively follows this survival probability and decreases for longer duration of the sensory phase. The longest sensory phase after which reactivation is no longer possible is well predicted by the time at which the excitatory strength (product of weight and number of synapses) drops below inhibition (*Figure 7C*, black line, *Figure 7F*, green region).

Within longer resting intervals, more synapses are created and stabilized, such that the stationary value increases (*Figure 7G*). Note, however, even a short rest phase of 3 min is sufficient to maintain strong connectivity for up to 20 hr of sensory phase.

## Role of short-term depression

In the above, spontaneous reactivations of assemblies were terminated by short-term depression. However, short-term depression is not the only candidate mechanism and spike-frequency adaptation can be used instead (*Jercog et al., 2017*). This does not lead to qualitative changes in the results thus far (data not shown), however, a qualitative difference emerges when considering the stability of overlapping cell assemblies. We initialized our network with two 30-neuron assemblies

with an overlap of 5 neurons, initially connected by 12 synapses on each connection. After 5 days, the network with short-term depression maintained the overlapping assemblies (*Figure 8A*). In contrast, the network with spike-frequency adaptation has formed non-overlapping assemblies (*Figure 8B*) and the neurons in the overlapping population have been incorporated into either one of the assemblies.

The reason for the difference is that with spike-frequency adaptation, neurons activated with the first assembly will be adapted and therefore less likely to be reactivated with second assembly. Due to this competition and the positive feedback between activity and connectivity introduced by the Hebbian plasticity processes, the overlap region will be reactivated and connected more and more with only one of the assemblies and disconnect from the other, separating the assemblies. In contrast, using short-term depression, only the synapses between the overlap region and the assembly are adapted such that a reactivation with another assembly is not impeded.

## Discussion

We have introduced a network with synaptic and structural plasticity which forms Hebbian cell assemblies in response to external stimulation. During random sensory stimulation these assemblies decay, but in the absence of external stimulation, the cell assemblies self-reactivate resulting in periods of strong correlated activity which strengthen and stabilize the intra-assembly connectivity, and weaken other connections. This protects the assemblies against ongoing synaptic turnover, increases their robustness against prolonged phases without reactivation, and leads to off-line improvement of the associative properties of the memories.

The critical ingredients and parameters for the mechanism to work are as follows: (1) The network needs to be able to spontaneously reactivate assemblies. This requires sufficient increase of the synaptic weights during the learning phase, so that the assembly has a net positive feedback and can transition to a high activity state. (2) The high activity state needs to terminate, requiring an adaptive mechanism such as spike frequency adaptation or short-term synaptic depression. (3) Lateral inhibition is required to prevent the activation from spreading to multiple assemblies. This de-correlates activity and ensures that only connections within the assemblies potentiate. (4) Structural plasticity should encourage stabilization of intra-assembly synapses. Here, this is achieved by a higher synapses removal rate for small synapses, while the synapse creation rate is fixed. Note, that the the structural plasticity in our model is not associative by itself. It becomes so indirectly via its dependence on the synaptic weights. (5) Connections between assemblies should remain weak, which, given (3) and (4), requires synaptic plasticity which depresses weights in case of asynchronous high pre- and postsynaptic activity, as is the case in Hebbian learning.

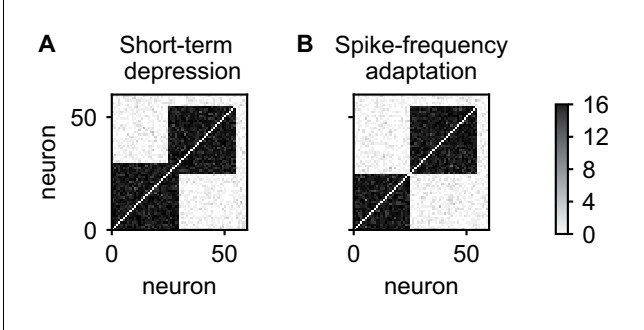

**Figure 8.** Stability of overlapping cell assemblies for alternative adaptation mechanisms. (**A**) Connectivity matrix of the number of synapses after 5 days for two overlapping assemblies and the combination of plasticity processes used in this paper. (**B**) Same for using spike-frequency adaptation instead of short-term depression. Neurons from the overlap-region have become associated to one of the assemblies and disconnected from the other. In all cases, connections within assemblies were initialized with eight synapses.

DOI: https://doi.org/10.7554/eLife.43717.011

## Self-organized reactivations

A number of earlier models of long-term and very-long-term memory have used reactivation to strengthen or restructure previously stored memories. However, these models often rely on an external reactivation mechanism (*Tetzlaff et al., 2013*; *Acker et al., 2018*). In contrast, here the self-generated reactivations of cell assemblies cause non-correlated reactivations of individual cell assemblies, such that neither individual external cues (*Acker et al., 2018*), nor pre-structured connectivity (*Tetzlaff et al., 2011*; *Tetzlaff et al., 2013*) are necessary.

Other models have shown that self-organized reactivation can stabilize Hebbian cell assemblies against ongoing synaptic plasticity (*Litwin-Kumar and Doiron, 2014*; *Zenke et al., 2015*). However, stability was only demonstrated on a timescale of hours and in the absence of synaptic turnover. It remained questionable whether assemblies could survive prolonged periods in which the reactivations are absent or when their synaptic substrate is subject to synaptic turnover. Here, we show that assemblies can indeed survive for many hours without reactivation. Moreover, in our model, reactivations maintain and strengthen the assembly connectivity by stabilizing new synapses which support the memories, while removing distracting synapses, resulting in offline memory gains.

We expect that offline memory gains will generally emerge for mechanisms which induce a bi-stable connectivity dynamics, that is when synapse creation balances removal both in a weakly and a strongly connected configuration and connectivity will converge to one of these configurations from any initial condition (*Figure 7D–G*). For example, the bi-stability does not necessarily have to rely on reactivation (see *Helias, 2008*; *Deger et al., 2012*; *Fauth et al., 2015*; *Deger et al., 2018* for alternatives). However, as connectivity dynamics in the presence of structural plasticity is slow, memory strength will typically not be saturated after learning. Therefore whenever the memory is strong enough, one should observe convergence to the upper configuration and hence a strengthening after learning. However, when only faster synaptic plasticity is considered for memory formation, the connectivity (i.e. the weights) tends to converge already during learning and no off-line strengthening will be observed (*Zenke et al., 2015*; *Litwin-Kumar and Doiron, 2014*; but see Figure 6 in *Tetzlaff et al., 2013* for an example of non-converged weights).

## Reactivation, sleep and memory improvement

Reactivations of memory related activity patterns in cortex have been reported mostly during sleep, predominantly during NREM or slow-wave sleep (*Ramanathan et al., 2015*; *Gulati et al., 2017*; *Gulati et al., 2014*; *Peyrache et al., 2009*; *Jiang et al., 2017*). During this sleep phase, the cortex exhibits alternating phases of high and low activity, so-called Up- and Down-phases or slow oscillations (*Steriade et al., 1993*; *Steriade et al., 2001*; *Timofeev et al., 2001*). This Up-Down-dynamics relies on recurrent excitation (*Sanchez-Vives and McCormick, 2000*), similar to the dynamics here. In vitro imaging of individual activity during Up-Down-dynamics in cortical slices reveal further similarities with our model: Each Up-state comprises a subgroup of cells with strong correlated activity and the time-course of the activity is reminiscent of the convergence into an attractive state (*Cossart et al., 2003*; *Shu et al., 2003*). Accordingly, also in vivo, Up-states induce stereotypical local patterns (*Luczak et al., 2007*). Finally, single-cell recordings indicate that learning-related activity patterns reactivate during slow-wave sleep and that this is phase-coupled with Up-states (*Ramanathan et al., 2015*). The memory strengthening reported here might therefore be related to the long known beneficial properties of sleep on memory (*Jenkins and Dallenbach, 1924*; *Karni et al., 1994*; *Fischer et al., 2002*; *Walker et al., 2003*; *Dudai, 2004*; *Stickgold, 2005*; *Gais et al., 2006*; *Korman et al., 2007*; *Lahl et al., 2008*; *Diekelmann and Born, 2010*; *Pan and Rickard, 2015*; *Rickard and Pan, 2017*).

In our model, we observe a decrease of memory stability for very long wake (sensory) phases (*Figure 7*). However, as sleep deprivation normally does not lead to a large scale loss of memories, it is possible that there are further processes at work to prevent the decay of memories. For example, there may be a stable pool of synapses (*Kasai et al., 2003*; *Loewenstein et al., 2015*), which is more resilient to decay. Another possibility is that reactivations do not exclusively occur during sleep, as sleep-like activity patterns can also be observed during periods of resting or quiet wakefulness (*Vyazovskiy et al., 2011*; *Sachidhanandam et al., 2013*; *Engel et al., 2016*; *Gentet et al., 2010*; *Poulet and Petersen, 2008*; *Okun et al., 2010*). Coherent with the above-presented model, offline-gains for some kinds of memories have been observed during wakefulness (*Walker et al.,*

*2003*; *Rickard et al., 2008*; *Cai and Rickard, 2009*; *Varga et al., 2014*; *Honma et al., 2015*; *Pan and Rickard, 2015*; *Rickard and Pan, 2017* but see, for example *Adi-Japha and Karni, 2016*). Moreover, several fMRI studies establish a link between the reactivation of task specific patterns during rest periods and later task performance (*Staresina et al., 2013*; *Deuker et al., 2013*), which is consistent with this view.

## Connectivity changes during sleep

The question whether cortical connectivity increases or decreases during sleep - especially slow-wave sleep - has been heavily debated recently (*Tononi and Cirelli, 2003*; *Frank, 2012*; *Frank, 2013*; *Tononi and Cirelli, 2014*; *Timofeev and Chauvette, 2017*). On one hand, the synaptic homeostasis hypothesis states that connectivity should be down-regulated during sleep to balance the potentiation dominated wake intervals and to allow for new learning (*Tononi and Cirelli, 2003*; *Tononi and Cirelli, 2014*). On the other hand, there is accumulating evidence that connectivity is built-up especially after learning (e.g. *Yang et al., 2014*, for a recent review see *Timofeev and Chauvette, 2017*).

Here, we observe an increase in the intra-assembly connectivity, consistent with the second hypothesis. Yet, inter-assembly and control connectivity decreases, as proposed by the synaptic homeostasis hypothesis. Thus, our model combines and refines both views: connectivity inside memory representations is up-regulated, such that memories are consolidated and strengthened, whereas the rest of of the network down-regulates weights and number of synapses, such that these neurons remain susceptible to subsequent learning.

## Synaptic weight fluctuations

While our model focuses on the critical ingredients described above, there are further challenges to the retention of memories on long timescales. Most prominent are the large intrinsic fluctuations of the synaptic weights observed on a daily basis (*Yasumatsu et al., 2008*; *Statman et al., 2014*; *Dvorkin and Ziv, 2016*). Although the daily changes of the synaptic weights in our model are comparable to experimental data, in the model the synapses segregate in a stable and unstable pool (*Figure 4—figure supplement 2*, compare to *Yasumatsu et al., 2008*, *Figure 1B*). It needs to be clarified in future research how such synapse intrinsic fluctuations affect memory maintenance. We expect that also in this context the connectivity build-up due self-reactivation will make memories more robust.

# Materials and methods

## Model description

### Neuron model

As the simulations extend to timescales of days and weeks, we use computationally efficient rate-based neurons. The membrane potential $u_i$ follows

$$\tau \frac{du_i}{dt} = -u_i + \sum_{j=1}^{N_{cells}} \sum_{k=1}^{S_{ij}} f_j w_{ij,k} v_j + I_{inh} + I_{stim,i} + I_{noise,i} \tag{1}$$

where $\tau = 155\mathrm{ms}$ is the neural time-constant, $I_{stim,i}$ is the individual external stimulation current and $I_{noise,i}$ is a spatio-temporal white noise current drawn from a Gaussian distribution with zero mean and standard deviation of 1.5. As neurons can be connected with multiple synapses, the connection between each pair $(i,j)$ of neurons is described by a number of synapses $S_{ij}$ and their synaptic weights $w_{ij,k}$ with $k \in \{1, ..., S_{ij}\}$ (see *Figure 1A*). The utilization factor $f_j$ arises from short-term depression (below).

Moreover, each neuron receives a global inhibitory current

$$\tau \frac{dI_{inh}}{dt} = -I_{inh} - w_{inh} \sum_{i=1}^{N_{cells}} v_i$$

determined by the sum of all firing rates $v_i$ and the inhibitory weight $w_{inh} = 3.5 w_{max}$, where $w_{max} = 0.7$ is a global factor scaling all weights (see below).

The firing rate follows from the membrane potential by a logistic function $v_i = (1 + \exp(-u_i))^{-1}$.

## Structural plasticity

We assume an all-to-all potential connectivity. Each pair of neurons has $S_{\max}$ potential synaptic locations at which functional synapses can be formed (*Figure 1A*). An unoccupied potential synapse is converted to a functional synapse with rate $b = 1\,\mathrm{day}^{-1}$. These new synapses are initialized at a small weight ($w_0 = 0.001$) and then evolve according to the synaptic plasticity rule described below. On the other hand, functional synapses will be removed with a certain probability. To model the experimental observation that larger synapses are more stable than small synapses (*Le Bé and Markram, 2006*; *Matsuzaki et al., 2001*; *Yasumatsu et al., 2008*), we use a deletion probability

$$d(w_{ij,k}) = d_1 + \frac{d_0 - d_1}{1 + \exp(-\beta(w_{off} - w_{ij,k}))} \tag{2}$$

which scales between $d_1 = 0.03\,day^{-1}$ (when $w_{ij,k} \to w_{max}$) and $d_0 = 24\,day^{-1}$ (for $w_{ij,k} \to 0$), with an offset $w_{off} = 0.35 w_{max}$ and steepness $\beta = 20$.

## Synaptic plasticity

The weights of existing synapses evolve according to a threshold-based Hebbian synaptic plasticity rule inspired by the calcium-based plasticity rule of *Graupner and Brunel (2012)*:

$$\frac{dw_{ij,k}}{dt} = \begin{cases} -\Delta_{decay} w_{ij,k} & \text{if } v_i < 0.5 \text{ and } v_j < 0.5 \\ +\Delta_{LTP}(w_{max} - w_{ij,k}) & \text{if } v_i > 0.5 \text{ and } v_j > 0.5 \\ -\Delta_{LTD} w_{ij,k} & \text{otherwise} \end{cases}$$

where $\Delta_{decay} = (2\,\mathrm{days})^{-1}$ is the weight-decay rate at low activity. The synapse potentiates (with a potentiation rate $\Delta_{LTP} = 0.1\mathrm{s}^{-1}$) when pre- and postsynaptic neurons are simultaneously highly active. Potentiation is soft-bound and diminishes close to the maximal weight $w_{max}$. When only one of the neurons is active, the synapse depresses with a rate $\Delta_{LTD} = 0.01\mathrm{s}^{-1}$. Note, this rule can also be seen as a variant of the covariance rule (*Sejnowski and Tesauro, 1989*) with decay instead of potentiation when both activities are low.

## Short-term depression

Similar as in previous models (*Holcman and Tsodyks, 2006*; *Barak and Tsodyks, 2007*), excitatory synapses are subject to short-term depression (*Tsodyks et al., 1998*; *Markram et al., 1998*; *Holcman and Tsodyks, 2006*; *Barak and Tsodyks, 2007*) to terminate high activty states. The utilization variable $f_i$ follows the presynaptic activity $v_i$:

$$\frac{df_i}{dt} = \frac{1 - f_i}{\tau_{relax}} - Ff_i v_i,$$

where $\tau_{relax} = 5\mathrm{s}$ is the time constant describing recovery from depression, and $F = 1\mathrm{s}^{-1}$ scales the amount of depression when the synapse is activated. Note, as the dynamics of the utilization variable only depends on the presynaptic activity, we can use the same $f_i$ for all synapses from the same presynaptic neuron $i$.

## Spike-frequency adaptation

We focus on short-term depression as a mechanism to terminate the high population activity. As an alternative mechanism (see, for example *Jercog et al., 2017*; *Setareh et al., 2017*), we use spike-frequency adaptation (*Benda and Herz, 2003*). We model this process as an additional current $I_{ad,i}$ in *Equation 1*. This current shifts the sigmoidal gain function and, thereby, adapts the firing frequency (*Jercog et al., 2017*). The adaptation current follows

$$\tau_{adapt} \frac{dI_{ad,i}}{dt} = -I_{ad,i} - \alpha v_i$$

with adaptation strength $\alpha = 33$ and an adaptation timescale $\tau_{adapt} = 5\mathrm{s}$ to achieve dynamics

comparable to short-term depression. In the simulations with spike-frequency adaptation, the utilization variable $f_i$ was fixed to 1 (no short-term depression).

## Simulations

We simulate a network of 240 neurons with an all-to-all potential connectivity of $S_{\max} = 16$ potential connections between each pair of neurons. The network is initialized without any functional synapses and exposed to a 6 hr long sensory phase (see below), during which structural plasticity converges to an equilibrium state in which synapse creation and removal are balanced. The subsequent simulation protocol has three phases (*Figure 1B*): During the learning phase, one of three disjunct stimulation groups of 30 neurons each receives a strong ($I_{stim} = 200$) stimulation for 18 s, which leads to nearly maximum activity in that group, followed by 36 s without stimulation. After that, the next group is stimulated. This protocol is repeated as long as the learning phase lasts (typically 9 h).

After learning, the network is alternately exposed to **sensory phases** and **rest phases** with durations drawn from exponential distributions with means $\tau_{\text{sens}} = 4\mathrm{h}$ and $\tau_{\text{rest}} = 2\mathrm{h}$ respectively unless stated otherwise. The sensory phase models the ongoing input from upstream networks arising from sensory information. During this phase, we randomly select 15 neurons and expose them to strong stimulation ($I_{stim} = 50$) for 1 s. After this, a new group of neurons is selected for stimulation. Note, we exclude the neurons that have been stimulated during the learning phase from the being stimulated in this phase. This guarantees that the random patterns do not (accidentally) reactivate the stored assemblies such that we are able to investigate the impact of phases without reactivation in our model.

Finally, during the rest phase, none of the neurons receives any external stimulation and activity is entirely intrinsically generated.

For simulations that investigate memory retention (*Figure 7B and C*, and *Figure 8*), we skip the learning phase and manually initialize strongly connected assemblies. Each connection within these assemblies starts with the same number of synapses with the maximum weight $w_{max}$, whereas all other connections start with no synapses. In these simulations, the sensory and rest phase durations are drawn from truncated normal distributions with a standard deviation set to 0.25 times the mean.

Simulations were written in C++ and optimized for efficient implementation of the above-described synaptic and structural plasticity rules on long timescales. For example, synapse creation and removal were simulated event-based. Differential equations were integrated with an Forward-Euler-method with a step-size of $100\mathrm{ms}$. Note that ideally the step size should be much smaller than the smallest time-constant in the system; however, this large value was chosen for efficiency and we checked that a shorter time-step did not substantially change the results (data not shown). Analysis was carried out in Python.

## Evaluating associative memory quality

To quantify pattern completion, we examine the currents evoked by corrupted versions of the learned activity patterns (stimulation groups). Corrupted pattern are created by randomly switching off a certain percentage of neurons which are active in the pattern , while the same number of inactive neurons is switched on. We evaluate the evoked current distributions in neurons which were active in the original pattern and in neurons that were inactive across 100 randomly corrupted versions of each pattern. For good pattern completion, these distributions should be well separated.

As performance measure, we evaluate how well the currents can be classified. Neurons which should be active should receive a current above the offset of the sigmoidal gain function in order to exhibit a high activity. Hence, to assess the influence of the corruption level, we abstract the neurons to binary units (i.e. active vs inactive classifiers) with a threshold at zero. We evaluate the percentage of neurons that reproduce the correct activity of the original pattern (again averaging over 100 corrupted versions of each pattern).

## Analysis of connectivity decay during sensory phases

To gain better insight into the mechanisms that influence the connectivity, we compare our simulations with an analytical theory. We first investigate the connectivity decay in the sensory phase during which the network receives random stimulation. For this, we use a mean-field approach and assume that an assembly is homogeneously connected with weights $w$ and a number of synapses $S$

between each pair of neurons (whose continuous value represents the expected number of synapses between two neurons). Due to the fast timescale of LTP, we assume that the weights have reached $w(0) = w_{max}$ at the end of a preceding learning or rest phase. During the sensory phase, the pre- and postsynaptic activity in the stimulated groups are almost always below the plasticity thresholds. Hence, weights will exponentially decay as $w(t) = w_{max} \exp(-\Delta_{decay} t)$.

Using the removal probability $d(w(t))$, *Equation 2*, the probability $s(t)$ that a pre-existing synapse survives until time $t$ is

$$s(t) = \exp\left[-\int_0^t d(w(\tau)) \mathrm{d}\tau\right] \tag{3}$$

$$= \exp\left[-\int_0^t d(w_{max}\exp(-\Delta_{decay}\tau))\,\mathrm{d}\tau\right], \tag{4}$$

which needs to be evaluated numerically due to the non-linearity of $d(w)$. Given a connection with $S(t=0) = S_0$ synapses, the expected number of surviving synapses is $S_0 s(t)$.

In addition to the surviving pre-existing synapses, new synapses will be created during the sensory phase at vacant potential synaptic locations with rate $b$. These synapses will remain at very small weights with a high removal probability $d \approx d_0$. We approximate their number $S_{small}$ by assuming that the creation and the removal of these small synapses is in equilibrium. Then, the expected number of created synapses equals the number of removed ones (*Deger et al., 2012*; *Fauth et al., 2015*), which gives

$$S_{small}(t) = \frac{b}{d_0 + b}(S_{\max} - S_0 s(t)).$$

The resulting time-course of the total number of synapses, $S(t) = S_0 s(t) + S_{small}(t)$, matches the connectivity decay in simulations (dashed line in *Figure 4B*).

## Analysis of maximal time without reactivation

To estimate the longest duration of a sensory phase after which an assembly can still self-reactivate, $t_{max}$, we determine the time-point at which the inhibition between the neurons within the assembly becomes stronger than the excitation. We neglect the $S_{small}$ newly formed synapses with small weights and assume that only $S_0 s(t)$ synapses with weights $w(t)$ contribute to the excitatory recurrent connectivity. Overcoming the inhibitory coupling between two neurons given by $w_{inh}$ requires $S_0 s(t) w(t) \geq w_{inh}$. We find the implicit relation for $t_{max}$

$$\exp(-\Delta_{decay} t_{max}) s(t_{max}) = \frac{w_{inh}}{w_{max} S_0}, \tag{5}$$

which is solved numerically to obtain an estimate of the longest time after which the assembly can still reactivate. Reactivation needs to happen before then, otherwise the connectivity keeps decaying and the assembly is lost.

## Analysis of one full cycle of sensory and rest phase

In the simulation, the network cycles between sensory and rest phases. We examine the expected connectivity changes within the stimulated groups for one full cycle of a sensory and rest phase. Until the end of the sensory phase, $t_{sens}$, the connectivity follows the above derived time-courses. Assuming that the assemblies self-reactivate during the rest phase, the surviving $S_0 s(t_{sens})$ synapses as well as the $S_{small}(t_{sens})$ small synapses formed during the sensory phase will quickly potentiate due to the correlated activity and remain stable throughout the rest phase. For longer resting interval durations $t_{rest}$, we also have to consider synapse creation during this phase. The probability that a vacant potential synapse is created during the rest phase is $p(t_{rest}) = 1 - \exp(-b t_{rest})$. Therefore,

$$\begin{aligned} S_{new} &= \underbrace{[S_{\max} - S_0 s(t_{sens}) - S_{small}(t_{sens})]}_{\text{vacant locations at } t_{rand}} \cdot [1 - \exp(-bt_{rest})] \\ &= \frac{d_0}{d_0+b}[S_{\max} - S_0 s(t_{sens})] \cdot [1 - \exp(-bt_{rest})] \end{aligned}$$

potential synaptic locations are turned into synapses and stabilized during the resting interval.

Hence, the expected change in the number of connections per full cycle is

$$\begin{aligned} \Delta S(S_0, t_{rest}, t_{sens}) &= S_{new} + S_{small}(t_{sens}) - S_0[1 - s(t_{sens})] \\ &= S_{\max}[1 - \gamma(t_{rest})] - S_0[1 - \gamma(t_{rest})s(t_{sens})] \end{aligned} \tag{6}$$

with $\gamma(t_{rest}) = \frac{d_0}{d_0+b}\exp(-bt_{rest})$.

To compare this to the simulation results (*Figure 7B*), we estimated the number of cycles in the simulation by dividing the simulation time of 5 days by the sum of the mean interval durations $\tau_{rest}$ and $\tau_{sens}$ and calculate the total expected change after the respective number of cycles.

Furthermore, we can use *Equation 6* to determine the stationary number of synapses $S^*$ at the start of a sensory phase for which we expect no change after one full cycle (i.e. $\Delta S(S^*, t_{rest}, t_{rand}) = 0$) as

$$S^* = S_{\max}\frac{1 - \gamma(t_{rest})}{1 - \gamma(t_{rest})s(t_{decay})}. \tag{7}$$

For this initial value, the number of synapses will still drop during sensory phases but will return to the same value after each full cycle. Whether this stationary state is stable and can be reached will be discussed below.

## Acknowledgements

MF was supported by German Research Foundation under project FA 1471/1–1. MvR was supported by the Engineering and Physical Sciences Research Council EPSRC EP/R030952/1.

## Additional information

### Competing interests

Mark CW van Rossum: Reviewing editor, *eLife*. The other author declares that no competing interests exist.

### Funding

| Funder | Grant reference number | Author |
| --- | --- | --- |
| Deutsche Forschungsgemeinschaft | FA 1471/1-1 | Michael Jan Fauth |
| Engineering and Physical Sciences Research Council | EP/R030952/1 | Mark CW van Rossum |
| Deutsche Forschungsgemeinschaft | FA 1471/2-1 | Michael Jan Fauth |

The funders had no role in study design, data collection and interpretation, or the decision to submit the work for publication.

### Author contributions

Michael Jan Fauth, Conceptualization, Formal analysis, Funding acquisition, Investigation, Visualization, Methodology, Writing—original draft, Writing—review and editing; Mark CW van Rossum, Conceptualization, Resources, Formal analysis, Supervision, Funding acquisition, Visualization, Methodology, Writing—original draft, Project administration, Writing—review and editing

Author ORCIDs
Michael Jan Fauth (iD) https://orcid.org/0000-0002-0732-2972
Mark CW van Rossum (iD) https://orcid.org/0000-0001-6525-6814

Decision letter and Author response
Decision letter https://doi.org/10.7554/eLife.43717.015
Author response https://doi.org/10.7554/eLife.43717.016

## Additional files

### Supplementary files
• Source code 1. Simulation source code.
DOI: https://doi.org/10.7554/eLife.43717.012

• Transparent reporting form
DOI: https://doi.org/10.7554/eLife.43717.013

### Data availability

All data generated or analysed during this study are included in the manuscript and supporting files. The source code zip archive (Source code 1) contains the model simulation code and the stimulation file used to generate Figure 2, 3, 4B&C, 5 and 6 as well as Figure 4—figure supplements 1 and 2.

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
