## [Decision Letter]

Thank you for submitting your article "Self-organized reactivation maintains and reinforces memories despite synaptic turnover" for consideration by *eLife*. Your article has been reviewed by two peer reviewers, and the evaluation has been overseen by a Reviewing Editor and Eve Marder as the Senior Editor. The reviewers have opted to remain anonymous.

The reviewers have discussed the reviews with one another and the Reviewing Editor has drafted this decision to help you prepare a revised submission.

Using computational models, the authors address the question of maintaining stable memories in the face of much synaptic perturbation. Both reviewers found the work an interesting expansion on previous studies, and intuitive and clearly written. However, questions were raised regarding (i) the necessity of synaptic turnover dynamics for assembly stability, and (ii) observations relative to experiment requiring some discussion. These aspects are detailed below and need to be addressed. Further, it would be appropriate to make public the model simulations used in the paper.

Essential revisions:

1) The authors have expanded on previous models (Litwin-Kumar and Doiron, 2014; Zenke et al., 2015) and have combined synaptic turnover and plasticity mechanisms to build stable assembly structure. These previous models only considered plasticity mechanics. Because of the long timescales involved in synapse creation and deletion the authors had to consider only firing rate dynamics (the past models used spiking neuron models), a much smaller network (hundreds instead of thousands), and a very large time step (100 ms) so that any fine timescale effects are ignored. Had they not made these simplifications the simulations would have taken an excessive amount of time on a reasonable computer.

However, both Litwin-Kumar et al. and Zenke et al. did produce stable assemblies without considering synaptic turnover, or at least they appeared stable for the simulation lengths they performed (hours – not days like the current manuscript). Also, they did have reduced mean field models that showed the assembly structure that was asymptotically stable, but these theories ignored any spiking dynamics so it is possible that the network simulations did not have stable structure (just a very slow decay).

I would like to know if the synaptic turnover dynamics in the current study are necessary for assembly stability. Basically, if the parameter *b* is set to zero in the equation for *d(w_ij,k_*) (subsection “Structural plasticity”) do assemblies now dissolve during the sensory and rest phases? Or does turnover just expand the region of parameter space where stable assemblies can be trained?

2) The paper addresses the question of synaptic turnover and strength modulation, specifically its effect on network functionality in terms of memory retention. In general this is a timely question that has received considerable attention recently, with several different possible resolutions to the problem of how stable functionality is maintained in face of synaptic fluctuations.

Here the authors investigate a computational model where memory relies on cell assemblies, and their stability is maintained by recurring spontaneous reactivation of the assemblies. The computational model is quite complicated and includes numerous processes: in addition to interaction among neurons, a global inhibition proportional to total firing rate is fed back to each neuron. External stimulus and external noise is added independently to each neuron, where the form of stimulus determines the particular protocol ('sensory', 'rest'). Three different forms of synaptic plasticity are included, spanning a broad range of timescales: LTP and LTD, on timescale ~10-100ms; short-term depression, ~1-5sec; and structural plasticity, ~hr-day, depending both on size and on firing rate (synapses tend to be deleted more when they are small, and/or when both neurons exhibit a low firing rate).

Part of the resolution to structural plasticity and synaptic turnover is found in the assumption of multiple potential synapses connecting each pair of neurons. This essentially adds a slower timescale to synaptic changes, and further stabilizes strong synapses by positive feedback. Thus while individual synapses turn over the pair can remained connected and neural assemblies remain stable.

Simulations of the model under the defined protocol – an extended period of embedding the memory patterns, followed by interspersed random inputs and rest periods – confirm and extend previous results on maintenance of assemblies through synaptic plasticity, over the longer timescales included in the model. Essentially the effect is a strong feedback internal to the formed assemblies, with lateral inhibition with the rest of the network, effectively forming a bistable system with reactivation by spontaneous firing. If the bistable system never goes below the transition point to the low firing state, it will easily be reactivated back to the high state. Learning is a very long process of embedding the memories in the system, utilizing a broad range of timescales (but still leaving some strengthening by reactivation by the slowest structural plasticity; "offline strengthening").

The paper is interesting and clearly written. Although the model is not simple, intuition is given for the observed effects. One disappointing feature is that the resulting synaptic fluctuations are far removed from those observed experimentally: they are essentially trapped at the highest value possible within assemblies, with small decreases between "rescues" back to their maximal value. This would merit at least some discussion.

---

## [Author Response]

Essential revisions:1) The authors have expanded on previous models (Litwin-Kumar and Doiron, 2014; Zenke et al., 2015) and have combined synaptic turnover and plasticity mechanisms to build stable assembly structure. These previous models only considered plasticity mechanics. Because of the long timescales involved in synapse creation and deletion the authors had to consider only firing rate dynamics (the past models used spiking neuron models), a much smaller network (hundreds instead of thousands), and a very large time step (100 ms) so that any fine timescale effects are ignored. Had they not made these simplifications the simulations would have taken an excessive amount of time on a reasonable computer.However, both Litwin-Kumar et al. and Zenke et al. did produce stable assemblies without considering synaptic turnover, or at least they appeared stable for the simulation lengths they performed (hours – not days like the current manuscript). Also, they did have reduced mean field models that showed the assembly structure that was asymptotically stable, but these theories ignored any spiking dynamics so it is possible that the network simulations did not have stable structure (just a very slow decay).I would like to know if the synaptic turnover dynamics in the current study are necessary for assembly stability. Basically, if the parameter b is set to zero in the equation for d(w_ij,k_) (subsection “Structural plasticity”) do assemblies now dissolve during the sensory and rest phases? Or does turnover just expand the region of parameter space where stable assemblies can be trained?

We thank the reviewer for this interesting suggestion. In response we have now carried out simulations where structural plasticity was turned off (Figure 4D). It shows that the structural plasticity helps to maintain the assemblies. It is important that the situation that we model is more challenging than modelled in the cited studies, where the network continuously reactivates spontaneously. The memories are strengthened during that phase in both these earlier studies and our study. However, during the sensory phase, when reactivation is suppressed, the memories decay. If these phases lasts long enough, memory will be lost which is now shown explicitly in Figure 4D. This confirms that the presence of structural plasticity make the assemblies less reliant on strong synaptic weights and more robust against longer and longer phases where reactivation are suppressed. In our opinion, this increased robustness as well as the better associative memory properties, which develop over time, are a qualitatively new effect, which has not been observed in these previous models.

We have clarified the manuscript on this matter.

2) The paper addresses the question of synaptic turnover and strength modulation, specifically its effect on network functionality in terms of memory retention. In general this is a timely question that has received considerable attention recently, with several different possible resolutions to the problem of how stable functionality is maintained in face of synaptic fluctuations. Here the authors investigate a computational model where memory relies on cell assemblies, and their stability is maintained by recurring spontaneous reactivation of the assemblies. The computational model is quite complicated and includes numerous processes: In addition to interaction among neurons, a global inhibition proportional to total firing rate is fed back to each neuron. External stimulus and external noise is added independently to each neuron, where the form of stimulus determines the particular protocol ('sensory', 'rest'). Three different forms of synaptic plasticity are included, spanning a broad range of timescales: LTP and LTD, on timescale ~10-100ms; short-term depression, ~1-5sec; and structural plasticity, ~hr-day, depending both on size and on firing rate (synapses tend to be deleted more when they are small, and/or when both neurons exhibit a low firing rate).Part of the resolution to structural plasticity and synaptic turnover is found in the assumption of multiple potential synapses connecting each pair of neurons. This essentially adds a slower timescale to synaptic changes, and further stabilizes strong synapses by positive feedback. Thus while individual synapses turn over the pair can remained connected and neural assemblies remain stable.

We note that the mechanism also works for single-synaptic connections. New synapses between any two neurons in the same assembly are strengthened and stabilized, no matter whether they are already connected or not, as all assembly neurons all take part in the reactivation.

Simulations of the model under the defined protocol – an extended period of embedding the memory patterns, followed by interspersed random inputs and rest periods – confirm and extend previous results on maintenance of assemblies through synaptic plasticity, over the longer timescales included in the model. Essentially the effect is a strong feedback internal to the formed assemblies, with lateral inhibition with the rest of the network, effectively forming a bistable system with reactivation by spontaneous firing. If the bistable system never goes below the transition point to the low firing state, it will easily be reactivated back to the high state. Learning is a very long process of embedding the memories in the system, utilizing a broad range of timescales (but still leaving some strengthening by reactivation by the slowest structural plasticity; "offline strengthening").The paper is interesting and clearly written. Although the model is not simple, intuition is given for the observed effects. One disappointing feature is that the resulting synaptic fluctuations are far removed from those observed experimentally: they are essentially trapped at the highest value possible within assemblies, with small decreases between "rescues" back to their maximal value. This would merit at least some discussion.

We thank the reviewer for raising this interesting point. We now evaluate how the weight dynamics in our model compares to the fluctuations of, e.g., the spine volume in Yasumatsu et al., 2008. For this, we evaluated the weight changes in time intervals of one day (as done in the experiments) starting at the first day after learning (t=24h). Interestingly, we find that the distribution of the relative changes is quite comparable to the Yasumatsu data (new Figure 4—figure supplement 2).

We furthermore included a new Discussion section on the intrinsic fluctuation of synapses.